# Characterisation of Single-Phase Fluid-Flow Heterogeneity Due to Localised Deformation in a Porous Rock Using Rapid Neutron Tomography

**DOI:** 10.3390/jimaging7120275

**Published:** 2021-12-14

**Authors:** Maddi Etxegarai, Erika Tudisco, Alessandro Tengattini, Gioacchino Viggiani, Nikolay Kardjilov, Stephen A. Hall

**Affiliations:** 1Univ. Grenoble Alpes, CNRS, Grenoble INP, 3SR, 38000 Grenoble, France; maddi.etxegarai@eurecat.org (M.E.); alessandro.tengattini@3sr-grenoble.fr (A.T.); cino.viggiani@3sr-grenoble.fr (G.V.); 2Division of Geotechnical Engineering, Lund University, 221 00 Lund, Sweden; 3Helmholtz-Zentrum Berlin (HZB), 14109 Berlin, Germany; kardjilov@helmholtz-berlin.de; 4Division of Solid Mechanics, Lund University, 221 00 Lund, Sweden; stephen.hall@solid.lth.se; 5Lund Institute of Advanced Neutron and X-ray Science (LINXS), 223 70 Lund, Sweden

**Keywords:** fluid velocity map, hydro-mechanical, fluid measurement, neutron imaging mechanics, saturated rock, heavy water, high-speed tomography

## Abstract

The behaviour of subsurface-reservoir porous rocks is a central topic in the resource engineering industry and has relevant applications in hydrocarbon, water production, and CO${}_{2}$ sequestration. One of the key open issues is the effect of deformation on the hydraulic properties of the host rock and, specifically, in saturated environments. This paper presents a novel full-field data set describing the hydro-mechanical properties of porous geomaterials through in situ neutron and X-ray tomography. The use of high-performance neutron imaging facilities such as CONRAD-2 (Helmholtz-Zentrum Berlin) allows the tracking of the fluid front in saturated samples, making use of the differential neutron contrast between “normal” water and heavy water. To quantify the local hydro-mechanical coupling, we applied a number of existing image analysis algorithms and developed an array of bespoke methods to track the water front and calculate the 3D speed maps. The experimental campaign performed revealed that the pressure-driven flow speed decreases, in saturated samples, in the presence of pre-existing low porosity heterogeneities and compactant shear-bands. Furthermore, the observed complex mechanical behaviour of the samples and the associated fluid flow highlight the necessity for 3D imaging and analysis.

## 1. Introduction

Heterogeneity due to localised deformation can significantly affect the permeability of rocks, giving rise to anisotropic flow and acting as barriers or conduits. Understanding the effect of such heterogeneities on fluid flow is key in optimising processes involving extraction or injection of fluids in geologic reservoirs, e.g., hydrocarbon and water production or CO${}_{2}$ sequestration, as well as in understanding natural processes, such as earthquake mechanisms.

At the laboratory scale, properties such as porosity and permeability have historically been measured as averages over a whole sample (bulk measurements), with point-wise sensors at the edges of the sample. However, this is clearly inadequate in the presence of strong heterogeneities, whether intrinsic to the material or strain-induced. Such heterogeneities can, for example, be characterised in terms of their structure using microscopy and tomography techniques in 2D or 3D, respectively [1]. It has been shown that the processes leading to deformation-induced heterogeneity, such as shear and compaction bands, can be characterised using time-lapse 3D imaging, e.g., X-ray tomography, and adapted image processing (e.g., Digital Volume Correlation, DVC) [2]. Furthermore, to study fluid transport in geomaterials, a number of authors have employed X-ray imaging [3,4,5] or positron emission tomography (PET) [6,7,8] with solute as tracers in the fluids.

Recent work has shown that neutron imaging is a useful tool for visualising fluid flow inside bulk objects such as rocks [9]. Unlike X-rays, neutrons are strongly attenuated by hydrogen, providing a good contrast of hydrogen-rich fluids in geomaterials [10]. Furthermore, different isotopes of an element can provide significantly different contrast to neutrons. For example, heavy water (D${}_{2}$O) has a much lower neutron attenuation than normal (“light”) water (H${}_{2}$O). This makes it possible to study fluid flow (e.g., H${}_{2}$O) into samples already saturated with a physico-chemically analogous fluid (e.g., D${}_{2}$O) to simulate a state close to single-phase flow. Additionally, several metals (e.g., aluminium and titanium) are relatively transparent to neutrons, allowing the use of pressure-confinement cells (e.g., hydrostatic or triaxial rock mechanics cells) for realistic in situ loading experiments on centimetre-size samples.

A number of studies have been presented involving neutron radiography to follow fluid flow in geomaterials, e.g., [11,12,13,14,15]. However, radiographic imaging can only capture some aspects of the heterogeneity of fluid flow, which, in heterogeneous materials such as rocks, can be pronouncedly three-dimensional. Masschaele [16] tracked the fluid movement in a 3D geomaterial for the first time using neutron tomography. However, the speed of image acquisition has been a limitation in the development of dynamic experiments in neutron tomography. Recent technical advances in neutron imaging instruments have enabled fast water infiltration into soils and the effect of plant-roots to be studied by high-speed neutron tomography, with imaging down to <10 s per tomography and 200 μm spatial resolution [17] at the CONRAD-II instrument at Helmholtz-Zentrum Berlin. New developments and higher neutron flux have allowed this time scale to be pushed down to about 1.5 s per tomography with the same spatial resolution at the NeXT imaging beam-line at the Institut Laue Langevin [18].

In this work, new neutron experimental protocols and 4D (3D in space plus time) image analysis methods are presented, aimed at gaining a greater understanding of the coupled hydro-mechanical behaviour in reservoir rocks (and with a wider applicability to other porous materials). A key novelty of the experimental work lies in the study of 3D, pressure-controlled water flow in already-saturated rock samples, enabling the study of (in first approximation) single-phase flow. Results from experiments on three samples of the same sandstone are presented, one intact, i.e., as cored from a larger sample, and two samples that have been deformed under controlled triaxial compression in the laboratory at different confinements. One of the deformed samples was, after analysis of the experimental data, found to contain a pre-existing, low-porosity, natural structure, whereas little evidence of new, localised deformation was apparent after the triaxial test. The other laboratory sample developed an inclined, largely compactive, shear band during the laboratory deformation. Consequently, the fluid flow properties in samples with three different deformation features have been investigated [19].

## 2. Experimental Method

To study heterogeneity in the intrinsic permeability induced by localised deformation, pressure-controlled flow tests were carried out. To study permeability within already saturated tests, the fluid front was detected thanks to the different attenuation of neutrons by heavy (D${}_{2}$O) and light water (H${}_{2}$O). While having nearly one order of magnitude difference in neutron opacity, these two fluids can, in first approximation, be assimilated from a physico-chemical standpoint.

### 2.1. Samples Studied

Three cylindrical samples of a Vosges sandstone, from the Woustviller quarry in the Vosges Mountains (France), were considered in this study. All the samples were 39 mm in diameter and 78 mm in height. This rock is composed of approximately 93% quartz, 5% microcline, 1% kaolinite and 1% micas. Quartz has a macroscopic attenuation coefficient for cold neutrons of 0.005 cm${}^{-1}$, i.e., a low overall attenuation of the neutron beam for the chosen sample size. The rock has a porosity of 22% and mean grain diameter of 300 μm. This rock has been studied previously by a number of authors, particularly in terms of its mechanical response, e.g., [20,21,22,23,24,25,26,27], its transport properties, e.g., [28], and also in previous neutron experiments, e.g., [14,29].

One sample was undeformed (VO01ME) and the other two samples were deformed dry under triaxial compression at 30 MPa (VO03ME) and 40 MPa (VO05ME) confining pressures, respectively. The apparatus used for the loading is detailed in [20,21]. The confining pressure and deviatoric pressure were controlled while the axial displacement and actual deviatoric pressure were measured. Based on previous results [22], the loading was stopped soon after the peak stress, as the purpose was to study the effects of early stage deformation on fluid flow, i.e., prior to significant evolution and notable opening of the expected shear bands. The deviator stress vs. axial strain curves for the deformed samples are shown in Figure 1 and are analogous for the two. However, the loading of the second sample was stopped at a higher level of axial strain and thus the shear band in this sample is expected to have a more developed deformation. To assess the deformation in the samples due to the triaxial testing, X-ray tomography images were acquired before and after the loading at 50 × 50 × 50 μm${}^{3}$ voxel size at Laboratoire 3SR using an X-ray CT scanner. The 3D strain fields were determined through Digital Volume Correlation (DVC) of these images to provide local quantification of the deformation in the samples for comparison with the fluid flow mapping.

### 2.2. Neutron Imaging Setup

The neutron tomographies presented here were acquired at the CONRAD-II beam-line, at Helmholtz Zentrum Berlin (HZB) ([30]) in the position more suitable for high-speed experiments. This position is closer to the end of the neutron guide, where the neutron flux is higher ($2\times 10^{8}$ n/cm${}^{2}$), enabling the acquisition of high-speed (1 min) tomography data sets. This gain in flux entails a loss of neutron collimation and, therefore, lower spatial resolution, since this is directly proportional to the distance between the sample and the source, *L*, and inversely proportional to the pinhole aperture, *D*. According to Banhart [31], the maximum blur, *d*, in a radiography can be approximated as d=l×D/L, where *l* is the distance between the detector and the sample. In this case, due to the thickness of the cell, the distance between the sample centre and the detector was 40 mm. Given an aperture of the pinhole of 30 mm, the L/D was 167 and the neutron penumbra (as defined by the blur, d) was 240 μm. To minimise the neutron activation of the device, the neutron beam was reduced to illuminate only the portion of the cell containing the region of interest. This region covered the sample itself and part of the top and bottom of the cell, resulting in a Field Of View (FOV) of 88×140 mm${}^{2}$. This FOV, which is slightly larger than the sample size, was necessary to monitor the position of the fluids in the system and thus determine the entry and exit times of the fluids in the sample, which is critical in the control of the experiments. An exposure time of 0.2 s for the neutron images was chosen as a compromise between signal-to-noise ratio and acquisition speed. A 4 × 4 binning was employed in the acquisition of the radiographs, resulting in a pixel size of 220 μm, which is consistent with the remaining contributing factors to the resolution. A total of 300 projections were acquired for each tomography, resulting in a total acquisition time for one tomography of 1 min. Due to the need for acceleration and deceleration of the rotation of the stage, the tomography data were acquired over 190${}^{\circ }$; the initial and final radiographs were discarded and just the projections from the central 180${}^{\circ }$ were used in the tomographic reconstruction.

### 2.3. Hydraulic Testing Setup

The setup designed for the hydraulic test in the neutron beam allows control of the flow rate, which is injected from the bottom of the fluid pressure at the top and of the confining pressure. A photograph and schematic of the setup mounted in the CONRAD-II instrument are shown in Figure 2. The system also allows for measuring fluid flow through the bottom and the top of the cell. The complete system consists of two water tanks, a water/air interface, six valves to control the water flow, a control box for the valves, two pressure regulators, a programmable syringe pump (model NE-4000, New Era Pump System), three pressure transducers, a displacement transducer, a National Instrument acquisition card, and the test cell. The pressure regulators and the syringe pumps are controlled during the experiment through a LabView interface, which is also used to monitor the pressure transducers and the displacement transducer. The test cell is a cylinder made up of 2017A aluminium alloy, which is bolted directly onto the rotation stage of the neutron instrument. It measures 172 mm in height and 74 mm outer diameter, with an aluminium wall of 7 mm thick. The design and drawings of the cell can be found in [19].

Two tanks store the fluids used during the tests. One tank contained the distilled water to be injected, while the other allowed the collection of the fluid that passed through the sample. The first water tank feeds the syringe pump connected to the bottom intake of the cell, which leads the water to the sample, while the second tank is connected to a water/air interface to control the pressure and measure the volume of water that comes out of the top of the cell. A water/air interface is a cylindrical container with two compartments divided by a membrane. The bottom space is full of air and connected to a pressure regulator, and the top space is full of water and connected to the flow system. Three pressure transducers measure the pressure as close as possible to the top and bottom of the sample, as well as the confining pressure. Finally, six electro-valves control the flow of the fluid. The hydraulic setup is shown schematically in Figure 2b.

The samples were saturated with heavy water prior to the tests inside a vacuum chamber with 5 mL of heavy water poured slowly into the chamber every 30 min. Before taking the samples to the HZB facility in Berlin, they were kept in the vacuum chamber for a week, then wrapped with teflon, covered with plastic film and vacuum–packed. The samples were placed in the cell, with the teflon wrapping, together with the porous stones that diffuse the flow coming from the hydraulic conduits. A thermo-restringent FEP (Fluoro–Ethylene–Propylene) membrane was shrunk with a heat-gun around the samples, porous stones and end caps to make the sample walls water-tight. The FEP membranes have very low neutron attenuation, although they are stiffer than membranes such as neoprene or latex. Therefore, prior to the heat shrinking of the FEP onto the sample, synthetic rubber was added around the base of the piston to prevent leakage. Moreover, two O-rings were placed around the membrane, at the top and bottom of the sample, to further improve the sealing.

Once the setup was mounted, the confining pressure was gradually increased to 100 kPa, from ambient pressure, while the top and bottom pore water pressures were raised to 50 kPa. Before starting the flow test, a small initial volume of light water was pushed through the system to fill the hydraulic conduits. This was applied faster than during the actual flow experiment to speed up the whole process, while radiographs were acquired to monitor the arrival of the light water. Once the light water reached the bottom of the sample, the flow was slowed down and the high-speed tomographies started. Due to the complex control of the setup, light water reached the bottom of the sample before the first high-speed tomography was acquired. However, the loss of data was found to be minimal, as all the strain localisation bands were located in the upper portions of the samples.

The flow rate for the three hydraulic tests was 4 mL/h. The flow in samples VO05ME, VO01ME and VO03ME was captured with, respectively, 336, 414 and 471 tomographies, which translates to 5, 7 and 8 h test durations. The acquisition of the images for sample VO05ME started once the light water was already in the sample, reducing the time needed for saturation.

## 3. Image Analysis

The reconstruction of the X-ray tomography data acquired before and after the mechanical loading was performed using the X-Act software (RX-Solutions, Chavanod, France). The resultant 3D images were analysed using the open source DVC software TomoWarp-II ([32]) to provide the strain fields of the tested samples. While the full strain tensor was determined, only the volumetric and shear strain fields are presented in the following. The X-ray tomography images were also used to characterise the intrinsic inhomogeneity of the samples that could affect the fluid flow. This included computation of the 3D porosity field using an in-house code developed at Laboratoire 3SR. The algorithm starts by separating the solid and fluid phases with a pre-defined greyscale threshold, followed by the counting of the number of voxels corresponding to each phase over a set of cubes centred on a regularly-spaced grid. The size of this measurement cube was adapted to find a proper elemental volume, which was found to be 30 × 30 × 30 voxels.

The neutron tomography data were reconstructed using the ASTRA reconstruction toolbox ([33,34]) assuming a parallel beam geometry and using the 3D SIRT GPU-based algorithm implementation ([35]). However, due to incoherent scattering and the polychromaticity of the neutron beam, lower than expected attenuation values are found in the centre of the sample, as well as higher attenuation values at the borders. Both these effects are well known and generally referred to as “beam hardening”. While corrections for this type of artefact exist, their correction in a dynamic imaging experiment, where the material is changing during the tomography acquisition, is not well developed and will require further improvements. From the vertical central slice of the reconstructed image of the VO01ME (Figure 3), it can be seen that the greyscale values are higher closer to the border of the sample (A) and decrease closer to the centre (D). These profiles also show that the advancing boundary (“front”) between the H${}_{2}$O and D${}_{2}$O filled regions of the sample is not a clearly defined step function, i.e., the boundary between the two fluids is not sharp in the images. This effect could be due to noise in the images, to the resolution relative to the front advancement during the rotation, which leads to blurring in the tomographic reconstruction, but can also indicate some degree of diffusion between the two fluids, as studied by Yehya in [36], or gradual mixing of the two fluids, as studied by Boon in [37]. However, such analysis of the transition between the fluids is beyond the scope of this study.

The output of the tomographic reconstruction is a collection (in this case of hundreds) of 3D volumes that represent, for each voxel, the attenuation of that subset of the sample to the neutron beam, represented by “greyscale values”, which can be related to the content of heavy or light water in the corresponding region of space. However, due to the incoherent neutron scattering, particularly by H${}_{2}$O, the initial greyscale value and, even more so, the final value (since H${}_{2}$O has a higher incoherent scattering cross section) of each voxel depends on the distance from the pixel to the centre of the sample. Therefore, the analysis approach proposed here evaluates the evolution of the greyscale for each voxel independently of the attenuation values in its surrounding neighbours.

To detect the moment when light water reaches a given voxel, a voxel greyscale is studied as a function of time (i.e., across all the tomographies). Given the existence of two distinct states (saturated with normal and heavy water) and a gradient between the two, the evolution of the greyscale value for a given voxel can be fitted by a sigmoidal function, i.e.,
(1)$$y=A+\frac{B-A}{1+{\left(C/x\right)}^{D}},$$
where *A* and *B* are the horizontal asymptotes of the function, and *C* represents the abscissa (i.e., the time, in this case) at which the greyscale is halfway between *A* and *B*, corresponding to the inflection point. Finally, *D* describes the maximum gradient of the curve. *A* and *B* are the greyscale values corresponding to a voxel saturated with heavy or light water, respectively. Since the transition from one liquid to the other is a complex and gradual process, as explained before, *C* is chosen to represent the arrival time of the normal water and *D* indicates the speed at which the light water replaces the heavy water in a voxel. Figure 1 is defined in the positive domain of x and y where x>0 and *y*ϵ(*A*,*B*).

Examples of the greyscale evolution at different voxels in the sample and the corresponding fitting are reported in Figure 4. All the voxels show an overall increase in attenuation through the test, corresponding to a higher content of light water, as well as, in all cases, an increase in the scatter in the values with increasing light water content.

As expected for the presented boundary conditions, H${}_{2}$O arrives earlier at the voxels at the bottom of the sample. The values of the lowest pixel are plotted in Figure 4a, and the C parameter of the sigmoidal fitting for this pixel is $C_{a}$ = 51.02, the minimum. The voxels presented in Figure 4b,c, which are higher in the sample, have $C_{b}$ = 173.78, and $C_{c}$ = 279.57. In Figure 4a, the initial plateau, representing voxels saturated with heavy water, is almost missing. Conversely, the final plateau, representing the voxel full of light water, is missing in Figure 4d. This shows the robustness of the method for different regions of the sample, even where there is little data for the initial or final stages of the flow. However, a better fitting could be achieved if more tomograms were acquired before the flow started and at the end of the test, which was not the case for these tests. To compensate for these lacking data and to stabilise the solution in some voxels, especially at the top and bottom of the sample, the first and the last three greyscale values are repeated to pad the data before fitting the sigmoidal function.

The voxels represented in Figure 4d–f are placed in the same horizontal slice, starting from closest to the boundary of the sample (Figure 4d) to almost its centre (Figure 4f) to investigate the effect on the fitting procedure as a function of the position inside the sample. The arrival times for each of the analysed voxels are: $C_{d}$ = 249.013, $C_{e}$ = 207.77 and $C_{f}$ = 189.65. The greyscale value of the voxels saturated with heavy water are similar in all three points ($A_{d}$ = 0.312, $A_{e}$ = 0.301, $A_{f}$ = 0.345). However, the B value varies more between the different cases ($B_{d}$ = 0.732, $B_{e}$ = 0.583, $B_{f}$ = 0.498). Thus, the increase in greyscale for the voxel close to the boundary, from the start to the end of the test (comparison of the *A* and *B* values) is 2.34 times, while for the voxel close to the centre, it is only 1.44 times. Moreover, for the voxel near the centre (f), the greyscale values in the final plateau are much more scattered. This example shows that the method of the sigmoidal fitting adapts to the beam hardening effect, enabling the analysis.

Fitting the evolution of the greyscale data for all the voxels over the full 4D data-set provides five 3D volumes, one for each of the fitting parameters, plus the error between the real data and the fitted curve. A vertical central slice and a horizontal slice of the fitting results for VO01ME are represented in Figure 5. The values of A varied from 0.23 to 0.41, whereas B varied from 0.40 to 1. This is coherent with the previous statement about the higher variation across the voxel greyscale values for voxels saturated with light water. The first high-speed tomography was acquired after the first arrival of the light water. Therefore, there is an absence of data in the bottom part, the black area. The early arrival of light water in the sample could also be the reason for the higher values of the A parameter in the bottom half of the sample. The third column shows the C parameter, the arrival time of the water, which is expected to vary from 0 to the maximum amount of tomographies (400 in this case). Some of the values exceed the maximum acquisition time. This could be due to a lack of proper light water saturation of the sample or to the relatively high noise. For the D parameter, the difference between the bottom and top of the sample could be related, as for the A parameter, to the early entrance of light water. Finally, the error map indicates the reliability of the fitting. The error can be seen to be higher in the central area, which is due to the lower signal-to-noise ratio, and in the top of the sample, as a result of the lack of full saturation.

To perform the experiments within the tight time schedule of the available beam-time, the water was not pushed at the same rate for the whole duration of the tests; the injection rate was slowed down as the H${}_{2}$O-D${}_{2}$O front traversed the areas of greatest interest (the shear band regions) and increased while the front passed through less deformed regions. Specifically, in the first 20 mm of the samples VO03ME and VO05ME, the flow was 50% faster. These changes in flow speed have been accounted for in the sigmoidal fitting by analysing the data in terms of acquisition time.

Even though all the fitting parameters of the sigmoidal function provide relevant information about the hydraulic process, the analysis presented here focuses on the C parameter. In particular, a 3D volume, where each voxel represents the arrival time (C value) at the corresponding position in space, has been computed for all the tests. The time-resolution of the arrival time, C, can be, in principle, better than the tomography time step, although the noise in the data and uncertainties in the fitting will affect this resolution. The central vertical slice of the 3D time map for sample VO01ME is presented in Figure 6a. It can be seen that there are a number of outlier values in the time map, which are related to poor sigmoidal fittings resulting from acquisition noise, but also, possibly, to the sample heterogeneity. Outliers defined by voxels with time values differing by more than ±50 min to the voxels within a 2-voxel radius have been removed and their values replaced by the mean of the 2-voxel radius neighbourhood. This outlier removal step was performed twice.

Apart from the outliers, artefacts also occur due to the ring artefacts in the reconstructed volumes, often caused by miscalibrated or “damaged” detector pixels. In the current data, the main ring artefacts are concentrated at the centre of the volumes and induce erroneously high arrival time values. These artefacts are, in general, not removed by the outlier-removal described above. Ring artefacts are best handled during the acquisition, but can be reduced in the reconstruction step, as has been done here, but some rings may remain. Further mitigation of ring artefacts is not considered here. A final step is a Gaussian filter of radius 5 voxels, applied twice. A representative result is shown in Figure 6b (to be compared to Figure 6a).

Time map volumes have been used to identify the fluid front for a series of arrival times, which are then used as input to determine the fluid flow speed using the approach described in [29]. This approach computes the speed for every voxel in the region between two consecutive fluid fronts from the sum of the euclidean distance of the voxel from each of the fronts. These voxel distances are then divided by the time step between the consecutive fronts to obtain the speed. Repeating the process for each consecutive pair of fronts and assembling the results provides a full flow-speed map volume covering the entire sample.

In [29], the front position in each tomography image is identified using a greyscale threshold to segment the dry and wet regions of the sample, since the contrast in greyscale between the regions was significant. The arrival time corresponding to each of these fronts was then given by the relative time of acquisition of the tomography. In the present work, similar binary images of fluid front have been produced, but starting from the arrival time maps and using thresholds on the arrival time to define the desired fluid front. This was needed because in the data presented here, the contrast in greyscale between the regions filled by heavy and light water is not clear, plus the threshold value between the regions varies with the position in the sample. The fitting of the sigmoidal function to the greyscale data in each voxel adapts to these spatial variations (including those due to the beam hardening effect). Furthermore, the sigmoidal-fitting provides a front-arrival time directly for each of the voxels in the 3D volume, virtually enabling a one-pixel resolution in the resulting speed map.

## 4. Results and Interpretation

In this section, results from the analysis, using the methods described above, are presented for the three samples tested: the intact sample, VO01ME and the two laboratory-deformed samples, VO03ME (30 MPa confinement) and VO05ME (40 MPa confinement).

### 4.1. VO01ME

Sample VO01ME is an “intact” specimen, i.e., cored out of a block of visually homogeneous Vosges sandstone, and has not been subjected to any additional mechanical loading in the laboratory. Therefore, stable Darcy-like flow and a homogeneous 3D flow speed map are expected.

Two perpendicular vertical central slices and the middle horizontal slice of the speed map are presented in Figure 7. Other than some heterogeneity in the flow speed near the top of the sample (likely reflecting some material heterogeneity), the flow is seen to be relatively constant over the sample height.

Based on the sample diameter of 39 mm and porosity of 22%, the total pore volume (and volume of fluid required to saturate the sample) is 19.48 mm, i.e., 0.2495 mL per millimetre height. Therefore, given the injection rate at the base of the sample of 4 mL/h, the vertical speed of the fluid advancement should be around 0.263 mm/min (1.19 voxels/min), which is in agreement with the measured speeds reported in Figure 7.

Despite the use of porous stones and an expansion chamber below the sample, the flow field suggests that there is a slight radial gradient of pressure at the base of the sample, which is at its highest above the injection point at the centre. This could be the cause of the low-curvature (about 1/50 mm${}^{-1}$) dome-shape of the flow front. Lower velocities near the sample edges are presumed to be related to viscous drag. However, this effect could be accentuated by an analysis artefact due to the curvature of the front, as discussed in [29].

The white patch at the bottom of the images is due to the arrival of the light water front before the start of the tomography acquisitions. Since in these areas no images were acquired in the initial condition, i.e., full of heavy water, the sigmoidal analysis cannot be performed. Conversely, the spurious high-speed values at the top of the sample are due to the lack of total light water saturation by the time the test was stopped, interfering with the sigmoidal fitting. Both these artefacts are caused by choices made to optimise the use of the limited beam-time. Another small artefact can be seen in the central region of about 2 mm radius, in the form of slightly higher velocities, which is likely caused by the residual ring artefacts that are more intense around the centre of rotation.

### 4.2. VO03ME

The triaxial compression test for sample VO03ME was stopped shortly after the peak stress in order to induce a partial shear band without allowing its full development. The shear strain values of this thin band (Figure 8b) are only slightly higher than the background level, in the order of two. The volumetric strain field shows that this region is slightly dilatant (Figure 8e), although overall the sample experienced compaction.

The 3D speed map in Figure 9 shows that the flow is essentially constant over the sample height and does not show any structure corresponding to the deformation seen in the strain fields. In fact, the overall behaviour is comparable to that of the intact sample VO01ME. However, significant heterogeneity is observed in the form of a vertically oriented plane where the flow is slower, e.g., to the right of the centre in Figure 9a and passing diagonally across the lower-right of the slice in Figure 9c. This structure cannot be correlated with the deformation seen in Figure 8 and was presumed to be related to pre-existing heterogeneity. Horizontal slices through the porosity map, from the X-ray tomography data acquired before the triaxial deformation, over the region of the sample exhibiting low flow speed, is shown in Figure 9c and Figure 10. A lower porosity structure can be observed with the same geometry as the flow heterogeneity, indicating that this is some pre-existing structure, likely related to either a natural compactant deformation and/or a reduction in porosity caused by, for example, a later cementation process.

### 4.3. VO05ME

The DVC analysis of the X-ray tomography data for sample VO05ME shows that the triaxial compression induced a shear band inclined at about 45${}^{\circ }$ with respect to the vertical axis, see Figure 11. Unlike in sample VO03ME, this shear band appears to be clearly formed. DVC-derived volumetric strain fields also show that the sample generally experienced compression, but some dilation can be seen in the shear band close to the specimen boundary.

The flow speed values in Figure 12 are consistent with those for the other samples, but the effect of strain localisation is easily distinguishable in Figure 12a,c as regions of reduced flow speed. Below the region with the shear band, the flow seems relatively consistent with the one observed in the other tests. Furthermore, as in the other two tests, regions of spuriously higher speed at the top of the sample can be seen due to incomplete light water saturation.

## 5. Discussion

In the previous section, results are shown correlating deformation structures with the speed of pressure-driven fluid flow in saturated samples of a sandstone. For the sample not subjected to deviatoric loading in the laboratory prior to the flow experiment (“as-cored”), relatively homogeneous flow is observed through most of the sample, with a flow speed consistent with Darcy-like flow. However, intrinsic inhomogeneities in the specimen appear to have led to local decreases in the speed of the fluid flow. A similar observation is also made for sample VO03ME; this sample exhibits a minor localised deformation feature induced by the in-laboratory triaxial loading (at 30 MPa confining pressure), but a more significant influence on the fluid-flow is induced by a pre-existing heterogeneity, which is interpreted to be a low-porosity, natural localised deformation band. For both of these samples, the flow front of the advancing light water into the heavy water is seen to have only a slight curvature, except near the boundaries of the samples where the flow speed is reduced, likely by the interaction between the flow and the boundary.

The third sample (VO05ME), which is deformed under triaxial compression at 40 MPa confining pressure in the laboratory, is seen (from DVC analysis of X-ray tomography data) to have developed a compactant shear band. Overall, the mechanical response (and also that of VO03ME) is in agreement with similar studies present in the literature [20,21,22,23,26]. However, previous studies have often shown a more dilatant behaviour in the shear band, whereas, in sample VO05ME, the band is mainly compactant. This could be linked to the fact that shearing was not allowed to fully develop in this sample, having been stopped right after the deviatoric-stress peak. Further deviatoric loading would likely lead to further displacement along the shear band and induce dilation.

During the flow test, the neutron imaging reveals that the compactant shear band feature in sample VO05ME acts as an obstacle to the fluid, as marked by the reduced flow speeds. This confirms the expectation of lower permeability in a compactant shear band, likely caused by the reduction in porosity and pore-throat size, as well as by the presence of fine, pore-filling material.

The results presented in this work represent the first observation in 3D of pressure-driven, single-phase flow in deformed rock specimens; previous, similar flow tests with 3D imaging are performed either with intact specimens or with capillary driven flow into dry specimens. The proposed experimental approach provides new insight into the influence of (localised) strain on permeability. It should be noted that only the speed field, and not the full velocity vector (speed plus direction), is currently measured. Future work should aim to quantify the full flow velocity. However, the method does provide an effective quantitative measurement of the front speed and, in doing so, enables the effect of heterogeneities, either natural or induced, on the fluid flow to be assessed, including unexpected ones (sample VO03ME). These effects can be directly related to local porosity or strains quantified through X-ray imaging. Future studies will extend this work to further explore the effect of different deformation features and link this to fluid flow.

In terms of the method presented in this work, the sigmoidal fitting approach enables improved analysis of the time-lapse 3D imaging data of the flow as it is robust to the varied evolution of the voxel greyscale in different parts of the sample, which can be influenced by a number of factors, including the flow field and imaging artefacts. The four-parameter function describes the transition from light to heavy water (and vice-versa) saturation of the voxels, adapting to the spatial differences in greyscale and being able to handle small and large changes in greyscale, due to the replacement of heavy by light water. The method can be easily adapted to both constant and varying fluid flow rates, adjusting the fitting to the imposed injection speed. This is particularly useful for experiments at large-scale facilities (which is typically the case for neutron imaging) where the allocated time for each test is limited. However, it is found that light-water should not be pushed into a test specimen before the acquisition of fast tomographies is started, and a test should not be finalised without a total saturation of the second liquid; both cases prevent the fitting of the data with the sigmoidal function.

The lower speeds observed at the boundaries of the samples, already observed in [29], are not fully explained by artefacts, in this case. In fact, the regions with slower flow are not symmetric, differ in each sample and were in this case significantly larger than in [29]. Although this could be due to a higher curvature of the fronts, it is possible that viscous drag at the boundaries of the sample is the source of the significant slowing of the flow observed. The beam-hardening artefacts that create higher greyscale values in the reconstructed neutron images close to the sample boundaries are reflected in the fluid arrival time map, e.g., Figure 5.

## 6. Conclusions

This paper presents an experimental approach to perform fluid injection tests to study single-phase flow with continuous 3D imaging by rapid neutron tomography, giving a 4D (3D plus time) view of the fluid advance in a sample. The developed equipment allows the control of the fluid-flow rate at the bottom of a sample and the outflow pressure at the top, while measuring the pressure at the bottom and the flow rate at the top. New 4D data analysis approaches are also presented, which enable 3D flow-speed maps to be determined from the 4D neutron imaging data. This approach can account for spatial variations in the image intensity and signal-to-noise ratios to provide robust measurements of the flow speed, as well as other aspects relating to the saturation and rate of saturation throughout a sample. Three samples of sandstone have been considered. One sample was intact, i.e., as cored from a larger sample, and two samples had been deformed under controlled triaxial compression in the laboratory, prior to flow testing. One of the deformed samples had a low imposed deformation (early inception of a shear band) and was, after analysis of the experimental data, found to contain a pre-existing, low-porosity, natural localised deformation layer. The other laboratory sample developed an inclined, largely compactive, shear band during the laboratory deviatoric loading. The experimental analysis reveals that variations in the flow speed maps correlate well with the observed localised deformation and pre-existing heterogeneities in the samples. In particular, the fluid flow is seen to be slower in laboratory-induced compactant shear bands, which is opposite to previous observations of accelerated flow under capillary-driven flow conditions of water into dry samples [14,29].

In addition to the direct observations for the studied samples, the proposed experimental approach and developed image processing tools open new avenues for the study of the complex hydro-mechanical interactions in porous media. In particular, this work paves the way for in situ mechanical loading (and hence deformation) combined with the fast acquisition of 3D images of the fluid advance. Together, such data will enable a better understanding of the effect of evolving, deformation-related heterogeneities on fluid flow.

## Figures and Tables

**Figure 1 jimaging-07-00275-f001:**
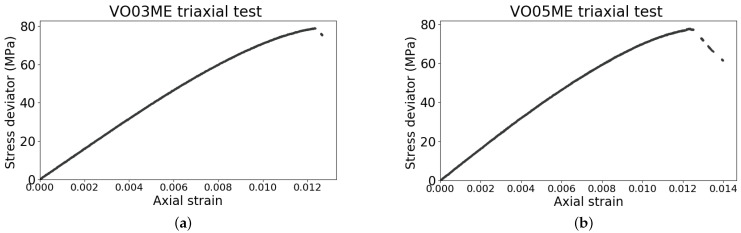
(**a**) Deviator stress vs. axial strain curve of the sample VO03ME at 30 MPa confining pressure. (**b**) Deviator stress vs. axial strain curve of the sample VO05ME at 40 MPa confining pressure. The discontinuities in the curves are due to the low acquisition frequency of the apparatus, 1 Hz.

**Figure 2 jimaging-07-00275-f002:**
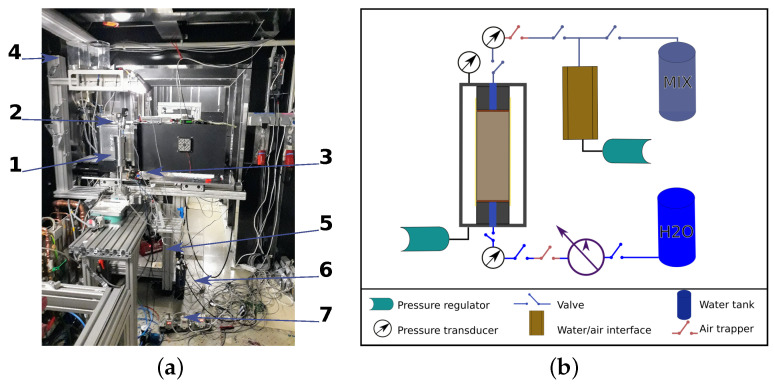
(**a**) Experimental setup inside the CONRAD II neutron beam. 1: Cell, 2: Top valve and pressure transducers, 3: bottom valves and pressure transducer, 4: water tanks, 5: pump, 6: water/air interface, 7: pressure transducers. (**b**) The hydraulic setup shows the tools connected to the cell during the test in order to control and monitor the fluid inside the sample and the pressure applied.

**Figure 3 jimaging-07-00275-f003:**
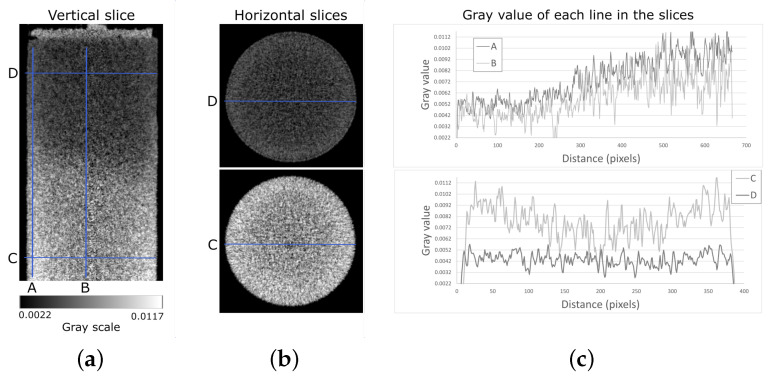
(**a**) Vertical slice of the VO01ME reconstruction at t = 100 min. The sample is saturated with heavy water, and light water is being pushed into the sample from the bottom. (**b**) Two horizontal slices of the reconstruction shown in (**a**). The slice of the top is saturated with heavy water and the bottom with light water. (**c**) Plots of the greyscale of the four lines drawn in images (**a**,**b**).

**Figure 4 jimaging-07-00275-f004:**
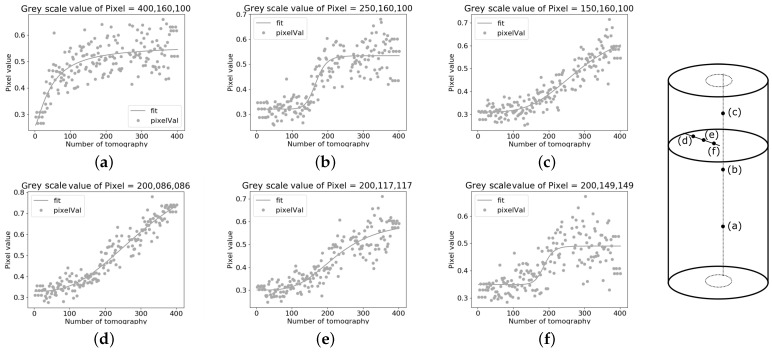
Sigmoidal fitting of the greyscale value of a voxel with time in sample VO01ME. Top plots: For position (160,100) in the horizontal different elevations. (**a**) 400 (**b**) 250 (**c**) 100. Bottom plots: For a given elevation, voxels with decreasing distance from the centre: (**d**) (86,86) (**e**) (117,117) (**f**) (149,149).

**Figure 5 jimaging-07-00275-f005:**
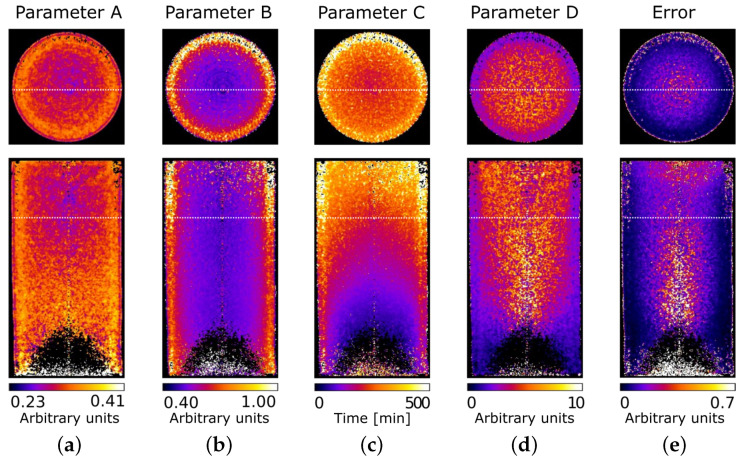
A vertical slice and a horizontal slice of the parameters obtained in the sigmoidal fitting of sample VO01ME (**a**) A, (**b**) B, (**c**) C, (**d**) D, (**e**) error.

**Figure 6 jimaging-07-00275-f006:**
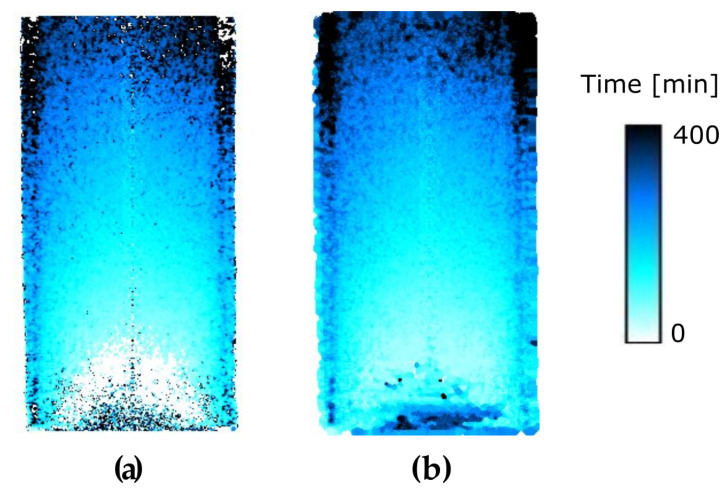
(**a**) Vertical central slice of the C parameter (representing the number of tomography at which the light water front arrives at a given pixel) of the sample VO01ME. (**b**) Same slice of the arrival time map after filtering.

**Figure 7 jimaging-07-00275-f007:**
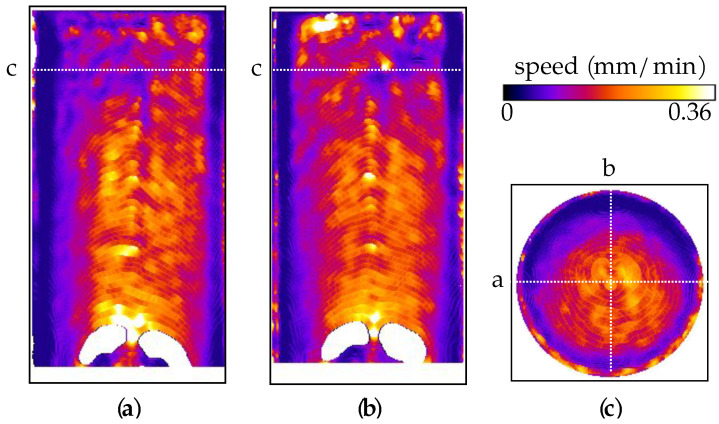
(**a**–**c**) Two vertical slices and a horizontal slice of the speed map of sample VO01ME.

**Figure 8 jimaging-07-00275-f008:**
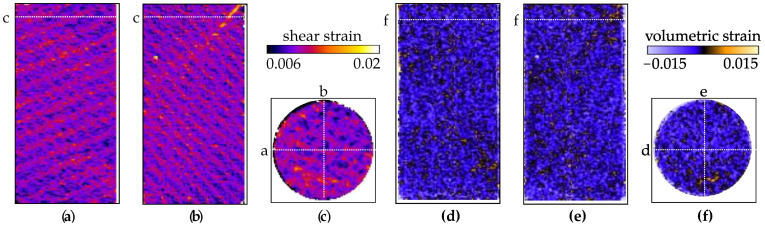
Two vertical slices and a horizontal slice of the Digital Volume Correlation for sample VO03ME. (**a**–**c**) Shear strain field. (**d**–**f**) Volumetric strain field.

**Figure 9 jimaging-07-00275-f009:**
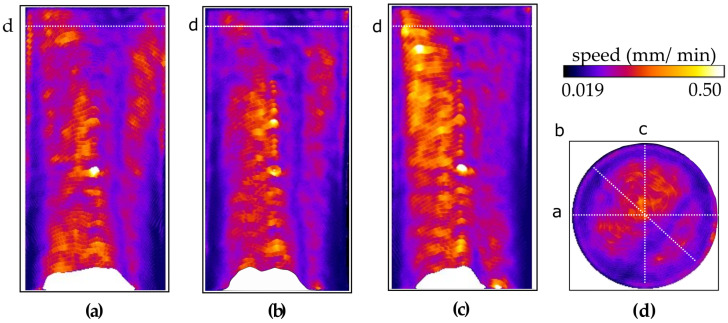
(**a**–**d**) Three vertical slices and a horizontal slice of the speed map from the flow test on sample VO03ME.

**Figure 10 jimaging-07-00275-f010:**
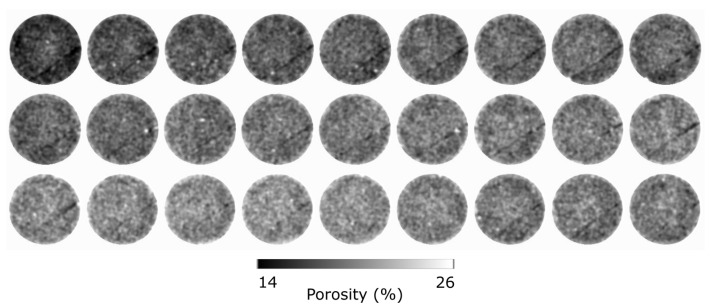
Twenty-seven equally separated horizontal slices along the top half of the porosity map of VO03ME.

**Figure 11 jimaging-07-00275-f011:**
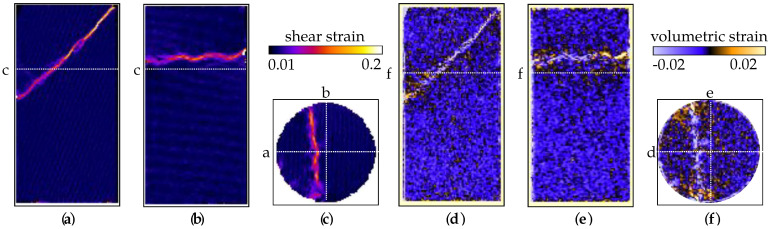
Two vertical slices and a horizontal slice through the DVC-derived strain fields for sample VO05ME. (**a**–**c**) Shear strain field. (**d**–**f**) Volumetric strain field.

**Figure 12 jimaging-07-00275-f012:**
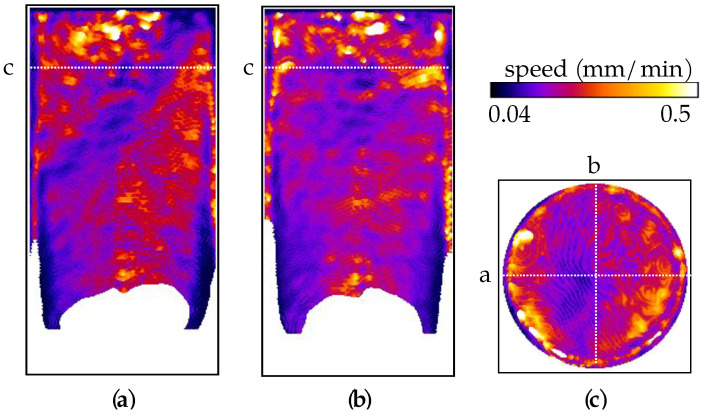
(**a**–**c**) Two vertical and an horizontal slice of the speed map of sample VO05ME.

## Data Availability

Not applicable.

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
