# Peer review of "Characterisation of Single-Phase Fluid-Flow Heterogeneity Due to Localised Deformation in a Porous Rock Using Rapid Neutron Tomography"

_2313-433X, 2021, doi:10.3390/jimaging7120275_

Round 1

Reviewer 1 Report

This study introduces the use of fast neutron tomography to track the velocity of a tracer concentration front in 3D as it is injected into a saturated rock sample. The methodology is novel and interesting, the presentation is good (with exception of the description of the flow set-up, which I found confusing), and I therefore think the paper can be published after minor revisions. My main comments relate to the fact that the experiments actually track a dispersive front, which includes mixing, rather than a "flow velocity". The two are closely related, but not exactly the same, even if diffusion can be neglected due to the time scale of the experiment. This is due to the occurence of mechanical dispersion (i.e. mixing of the two fluids due to the occurence of different flow velocities inside and between different pores). This affects the width of the sigmoidal fits, and depending on the image characteristics may also introduce a bias in the velocity determination in different parts of the sample. The authors should acknowledge this and perhaps see if they can extract interesting information on the mixing itself from their data. Related to this: solute dispersion has been studied using medical CT imaging and PET-imaging at similar temporal and spatial scales to the current study (e.g.: DOI 10.1002/2014WR015351, DOI 10.1002/2016WR019912, DOI 10.1016/j.advwatres.2019.03.003, and references therein). Some of these methods may be applicable to as alternatives to the current approach to the scientific question investigated in this paper. At a smaller sample scale (using micro-CT), DOI 10.1029/2019WR025880 even used a similar approach with a sigmoidal fit to the tracer concentration in individual pores. Given the framing of this paper as the presentation of a new methodology, the authors should treat these methods in the introduction and compare the them in terms of both practicality and information that can be gained.

Minor comments:

L34: "ideal" tool; this depends on the scientific question, please add more nuance.
L54: please add an indication of the spatial resolution obtained for these time resolutions.
L71: what's the relevance of the fact that this is part of a larger study to the reader? Rephrase or remove.
L138: this list should be made more relevant to the reader, for example: I don't need to know you have 3 power supplies because this is inconsequential to the results, but I would like to know more about the flow cell design and the syringe pump you used (ie details that may actually affect the results).
L149: "connected to a water air interface". this is confusing. do you mean you change the back pressure by putting the water reservoir at a certain height?
L153: I don't really see why you need 6 valves for this experiment, several of your valves seem to be in series. Please introduce the flow setup more clearly.
L158: vacuum packing seems risky to remove some of the water from the core. My experience is that in-situ saturation (or resaturation) works better. See also my next point.
L170: Where does the flushed air go? do you pump it out through the sample (this works if you can flush fast enough)?
L179: please report the fluid flux (more relevant to field scale) and the Reynolds and Péclet numbers of the flow.
Figure 3: please add in the caption that this is neutron data rather than X-ray data, for clarity.
Eq1: I suggest to change the names of the symbols to make their meaning more intuitive (eg change C by T_a; A and B by G_LW and G_HW; ...)
L283: this will influence (increase) the dispersion.
L355: the difference in viscous drag in a pore at the sample boundary and a pore in the center should be negligible and in any case would not cause a decay in flow rates on the observed length scale, so I really don't see this as a plausible explanation. More likely this is due to image artifacts and potentially the H20 vs D20 concentration profile at the inlet (due to dispersion when you bring the front to the bottom of the sample), which is then changing shape throughout the experiment.

Reviewer 2 Report

This study investigated the characteristics of single-phase fluid-flow heterogeneity due to localised deformation in a porous rock using rapid neutron tomography. Some minor revisions can be considered to improve the manuscript.

1.In figure 1, the curves after stress deviator peak are not continuous. Why?

2.In figures 7 and 9, there are some white zones without data. Please explain the reason.

3.The conclusions should be shortened.

Round 2

Reviewer 1 Report

I am okay with publication of the manuscript in its current form (even though I would have hoped for a little bit more context in the comparison of the current method with the X-ray and PET-based approaches).